# Quantum dissipation driven by electron transfer within a single molecule investigated with atomic force microscopy

Jan Berger[1,5], Martin Ondráček [1,5], Oleksandr Stetsovych[1,5], Pavel Malý[2], Petr Holý[3], Jiří Rybáček[3], Martin Švec[1,4], Irena G. Stará [3], Tomáš Mančal [2], Ivo Starý[3] & Pavel Jelínek [1,4 ✉]

Intramolecular charge transfer processes play an important role in many biological, chemical and physical processes including photosynthesis, redox chemical reactions and electron transfer in molecular electronics. These charge transfer processes are frequently influenced by the dynamics of their molecular or atomic environments, and they are accompanied with energy dissipation into this environment. The detailed understanding of such processes is fundamental for their control and possible exploitation in future technological applications. Most of the experimental studies of the intramolecular charge transfer processes so far have been carried out using time-resolved optical spectroscopies on large molecular ensembles. This hampers detailed understanding of the charge transfer on the single molecular level. Here we build upon the recent progress in scanning probe microscopy, and demonstrate the control of mixed valence state. We report observation of single electron transfer between two ferrocene redox centers within a single molecule and the detection of energy dissipation associated with the single electron transfer.

[1] Institute of Physics of the Czech Academy of Sciences, Cukrovarnická 10, 16200 Prague 6, Czech Republic. [2] Charles University, Faculty of Mathematics and Physics, Ke Karlovu 5, 121 16 Prague 2, Czech Republic. [3] Institute of Organic Chemistry and Biochemistry, Czech Academy of Sciences, Flemingovo nám. 2, 16610 Prague 6, Czech Republic. [4] Regional Centre of Advanced Technologies and Materials, Palacký University, Šlechtitelů 27, 78371 Olomouc, Czech Republic. [5] These authors contributed equally: Jan Berger, Martin Ondráček, Oleksandr Stetsovych. ✉email: jelinekp@fzu.cz

nvestigation of the interplay between charge transfer and dissipation on the molecular scale has a long history due to the importance of these phenomena in many biological, physical, and chemical processes, as well as for the design of functional molecular devices[1–5]. On an ensemble level, the role of dissipation in single-electron transfer is well understood. However, studies of electron transfer and the accompanying energy dissipation processes in a single molecule still represent a challenge[6,7]. Spectroscopic measurements employing short laser pulses provide unprecedented time resolution, but they typically require large ensembles of molecules[8,9], which makes the investigation of charge transfer and dissipation processes on the single-molecular level difficult. On the other hand, recent progress in frequency modulated atomic force microscopy (AFM)[10] enables us to control single-electron charge states in quantum dots[11], atoms[12], and molecules[13]. The technique has also demonstrated a possibility to drive intermolecular charge transfer[14], to detect the reorganization of the energy of a single molecule on a substrate[15] or to transduce the tunneling current flowing through a single molecule to the motion of a macroscopic oscillator[16] or orbital image[17].

In biological systems, charge transfer figures in various key processes, from energy transduction to $CO_2$ reduction and water splitting, to nucleic acid biosynthesis. Interestingly, the character of charge transfer is quite different in proteins and in nucleic acids such as DNA[18]. In proteins, the charges are largely localized and the transfer is directional as the charge transfer occurs over distant cofactors and substantially large energy gaps. In contrast, in nucleic acids, the distances and energy barriers are much smaller, resulting in fast, near-isoenergetic, coherently-delocalized transfer. Recently, the mechanism of flickering resonance in charge transfer has been described in DNA and other biological molecules, outlining the dynamic nature of these systems[19]. In this process, the fluctuations of the molecular structure lead to change in the energies of the electronic states, bringing them transiently into resonance. This transient near-degeneracy enables coherent charge transfer with signatures similar to tunneling, possibly accelerating biophysical reactions. The dynamic character of the environment influencing the electron transfer processes highlights phenomena such as pathway interference, conformational transfer gating, or the already mentioned transient resonances[20] in which the traditional picture of a static molecular structure must be replaced by a more dynamic one.

Our goal in this work is to extend the outstanding capabilities of the scanning probe microscopy (SPM) technique to controlling mixed-valence states and electron transfer within a single molecule. We demonstrate that we can indeed use the atomic force microscope (AFM) to create and probe mixed-valence state[21] of a molecule. The sensitivity of AFM to the mixed-valence state is tightly related to electron transfer between different redox centers within a single molecule, induced by the oscillating probe of the microscope. The AFM tip oscillations create a situation very similar to the transient resonance induced otherwise by thermal motion[3], allowing us, however, to manipulate the charged molecules and the conditions of the energy transfer in a controlled way. What is more, AFM also allows us to detect underlying energy dissipation to the environment. The back action of single electron tunneling between two redox centers on an oscillating scanning probe allows us to determine both the average switching rate[22] and dissipated energy as a function of temperature. We find the dependence of the switching rate on temperature to be negligible. Our simulations with environment reorganizing both fast and slow with respect to external driving modulation demonstrate that continuous intramolecular electron transfer drives the electron out of its thermal equilibrium. We show that the dissipation signal provides information about the character of the electron transfer dynamics. The weak temperature dependence of the switching rate is solely due to the non-equilibrium character of the electron state.

We believe that our work opens new possibilities for studying mixed-valence state molecular systems and quantum dissipation out of equilibrium on the single-molecular level using AFM, simulating in a controlled way key features of the dynamic environment that the molecular systems often experience. Strong coupling between the single-electron transfer within a single molecule and the dynamics of the macroscopic probe may also define novel single-molecular nanoelectromechanical systems[23] toward applications as molecular quantum-dot cellular automata[24–27].

## Results

**Control of single-electron charge states by means of AFM.** We report low temperature ultra-high vacuum AFM measurements of electron transfer within the 2,6-bis(ferrocenyl)naphthalene

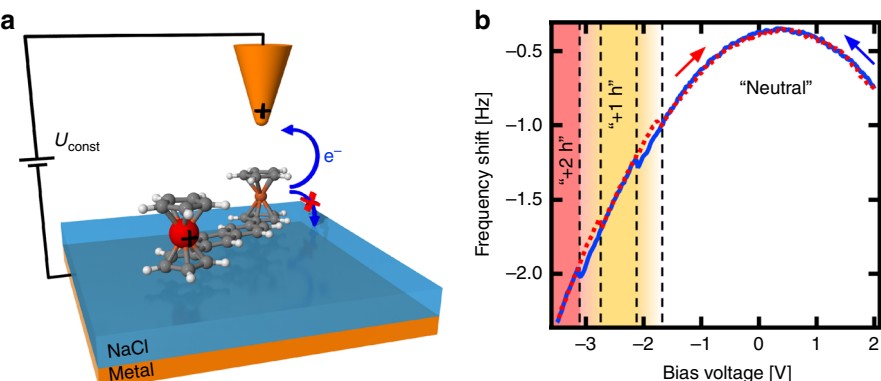

**Fig. 1 Charging of the molecule. a** Schematic view of the experimental setup, showing the definition of the sample voltage $U$, transport of electrons into the tip as well as within an adsorbed molecule. Thick layer of NaCl prevents tunneling current between bisFc molecules and the metallic substrate. **b** Charging parabola (blue solid line) taken above a bisFc molecule. Jumps in the parabola denote the extraction of an electron from the bisFc molecule, which brings the molecule to the once ($U = -2.1$ V) and twice ($U = -3.1$ V) positively charged states (light orange and light red regions). Discharging parabola (red dotted line) jumps are shifted in comparison with charging ones, creating hysteresis of charging. Jumps in parabola denote insertion of an electron to the bisFc molecule, which brings the molecule to the once positively charged ($U = -2.8$ V) and neutral states ($U = -1.8$ V).

(bisFc) molecule which consists of two ferrocene redox centers separated by a naphthalene linker. We deliberately selected a molecule with ferrocene units, which may undergo a reversible oxidation process from ferrocene (Fe$^{II}$) to ferrocenium salt (Fe$^{III}$)[28]. Each of these redox centers, mostly located on the iron atom having a $d$-like orbital character, may accommodate a single-electron charge. The presence of the naphthalene unit ensures sufficient localization of each redox state on a Fc center. As such, the bisFc molecule represents a realization of a two-level quantum system, which has great importance in many areas of quantum physics[29].

We deposited bisFc molecules on sufficiently thick NaCl (>10 monolayers (ML)) grown on an Ag(111) surface to prevent additional electron tunneling between the molecule and the metallic substrate, see Fig. 1a. As a consequence, we do not detect any tunneling current signal between the tip and the sample in the whole range of the applied bias voltages during our experiments. After deposition, we found individual molecules on the surface with two ferrocene units in a slightly different register with the substrate, see Supplementary Fig. 1. This is caused by the incommensurable length of the molecule with respect to the NaCl substrate. According to the density functional theory (DFT) calculations, the naphthalene unit adopts a planar configuration with two ferrocene units in the upright position, as shown in Fig. 1a.

Besides the frequency shift $\Delta f$ (gradient of force acting between tip and sample), we also monitor the excitation energy $E_{diss}$ during the experiment. The excitation energy $E_{diss}$ represents the energy loss of the sensor during one oscillation cycle, caused by the presence of non-conservative forces acting on the sensor[30]. This quantity allows us to monitor dissipation processes in the molecular system.

By positioning the AFM tip above the bisFc molecules, as schematically shown in Fig. 1a, and sweeping negative sample bias voltage $U$ (i.e. a positive charge on the tip), we can reversibly switch the molecule from neutral to single-hole-charge or double-hole-charge positively charged states. This occurs by withdrawing one or two electrons, respectively, from the redox centers of bisFc to the metallic tip and vice versa. The formation of these single (+1 h) and double (+2 h) oxidized charge states is revealed by characteristic jumps in the frequency shift $\Delta f$ in Kelvin $\Delta f(U)$ parabola, see Fig. 1b, which are similar to previous works[14,31,32]. The exact shape of the Kelvin parabola and the bias voltage at which such characteristic jumps occur depend on the tip properties, as already discussed by Steurer et al.[14].

In particular, the possibility to form a long-living mixed-valence state[21], when only a single electron is withdrawn from the system, is appealing. The ferrocene unit from which the electron was removed is converted into a bare ferrocenium Fe$^{III}$ cation, while the second ferrocene unit of the same bisFc molecule remains neutral in the Fe$^{II}$ state. This allows us to prepare a mixed-valence state in a single molecule by means of AFM. Note that the thick insulator layer preventing the electron tunneling to the substrate enables a long lifetime (several days) of the mixed states. In the following, we show that we can control the localization of the +1 h charge state on one of the Fc centers by the position of the tip, which acts as a gate.

Figure 2 displays simultaneously recorded constant-height images ($\Delta f$ and $E_{diss}$) acquired over the same molecule at the same tip-sample distance for three different bias voltages corresponding to neutral, single (+1 h) and double (+2 h) charge states, respectively. Interestingly, we observe quite a distinct contrast for the mixed-valence state (+1 h) with respect to the neutral and fully oxidized (+2 h) charge states. Namely, we detected a strong signal that appears as a sharp line in both $\Delta f$ and $E_{diss}$ channels in between the two redox centers in the mixed-valence state (see

Fig. 2b, e). This signal is missing in the case of the neutral and the fully oxidized (+2 h) charge states, Fig. 2a, c, d, f.

The presence of the charge state changes the frequency shift contrast of the bisFc molecule, as it can be seen from Supplementary Fig. 2. We should note that in far tip-sample distances, we cannot distinguish distinct molecular charge states in the $\Delta f(U)$ curves anymore. This is demonstrated by a series of $\Delta f(U)$ parabolas plotted in Supplementary Fig. 3. On the other hand, we can still observe the sharp line in both $\Delta f$ and $E_{diss}$ channels in between the two redox centers in the mixed-valence state at elevated tip-sample distance even at low bias voltages, see Supplementary Fig. 4.

An additional 2D $\Delta f$ x-z spectroscopy along the direction connecting the two redox centers reveals a parabolic extension of the sharp line signal out of the surface, as shown in Fig. 2g, h. This characteristic line is always present when we scan the system in a bias voltage range corresponding to the mixed-valence state. The shape and magnitude of the line signal depends on the particular tip apex shape (see Supplementary Fig. 5) but it is almost unaffected by variation of the bias voltage in the range corresponding to the mixed-valence state. In particular, we have seen that the shape of the line changes significantly when we intentionally picked some material from the ionic NaCl surface by the tip. This indicates that the shape is determined by the electrostatic interaction between permanent (rather than bias-induced) charges on the tip apex and the sample, as confirmed by our numerical simulations (see discussion below and Supplementary Fig. 6).

Considering that we only observe the singularity line in the $\Delta f$ and $E_{diss}$ channels for the +1 h charge states, we attribute its origin to a continuous switching of the charge between the two redox centers at a certain tip-sample position with respect to the redox centers.

Here, the AFM probe acts as an external gate, the position of which allows us to modulate the energies of the two-level system via electrostatic interaction with a charged redox state. When a single electron is withdrawn from a redox center of the neutral molecule to the tip located above this redox center, the repulsive interaction between the positively charged tip apex and the newly created hole causes transfer of the hole to the other, more distant, ferrocene unit, thus reducing the electrostatic interaction. Under such conditions, the positive charge remains fully localized on the distant redox center, see Fig. 3a. However, as the tip scans across the molecule, the electrostatic interaction with the AFM probe is changing accordingly. At a certain tip position, the configuration energies of the +1 h charge located on one or the other redox center become equal, and the charge may switch back and forth between the two redox centers, see Fig. 3b.

On top of this, the oscillating AFM probe is continuously infringing and reestablishing this balance during a single oscillation cycle. Importantly, the presence of the dissipation signal reveals that the switching mechanism occurs out of phase with respect to the AFM probe oscillation[11,30,33]. The discontinuity of the $\Delta f$ signal indicates that there is a difference of electrostatic forces acting on the AFM tip, depending on whether the charge is located on the left or on the right redox center. The dissipation signal reveals presence of non-conservative forces during the oscillation cycle caused by the charge transfer between two redox centers. Such non-conservative force component arises whenever the transfer is out of phase with the tip oscillation. It means that the probe senses a slightly different force when the +1 h charge states sits on distinct redox centers. The difference can be attributed to different screening of the +1 h charge state or different atomic relaxation of both the molecule and underlying substrate on

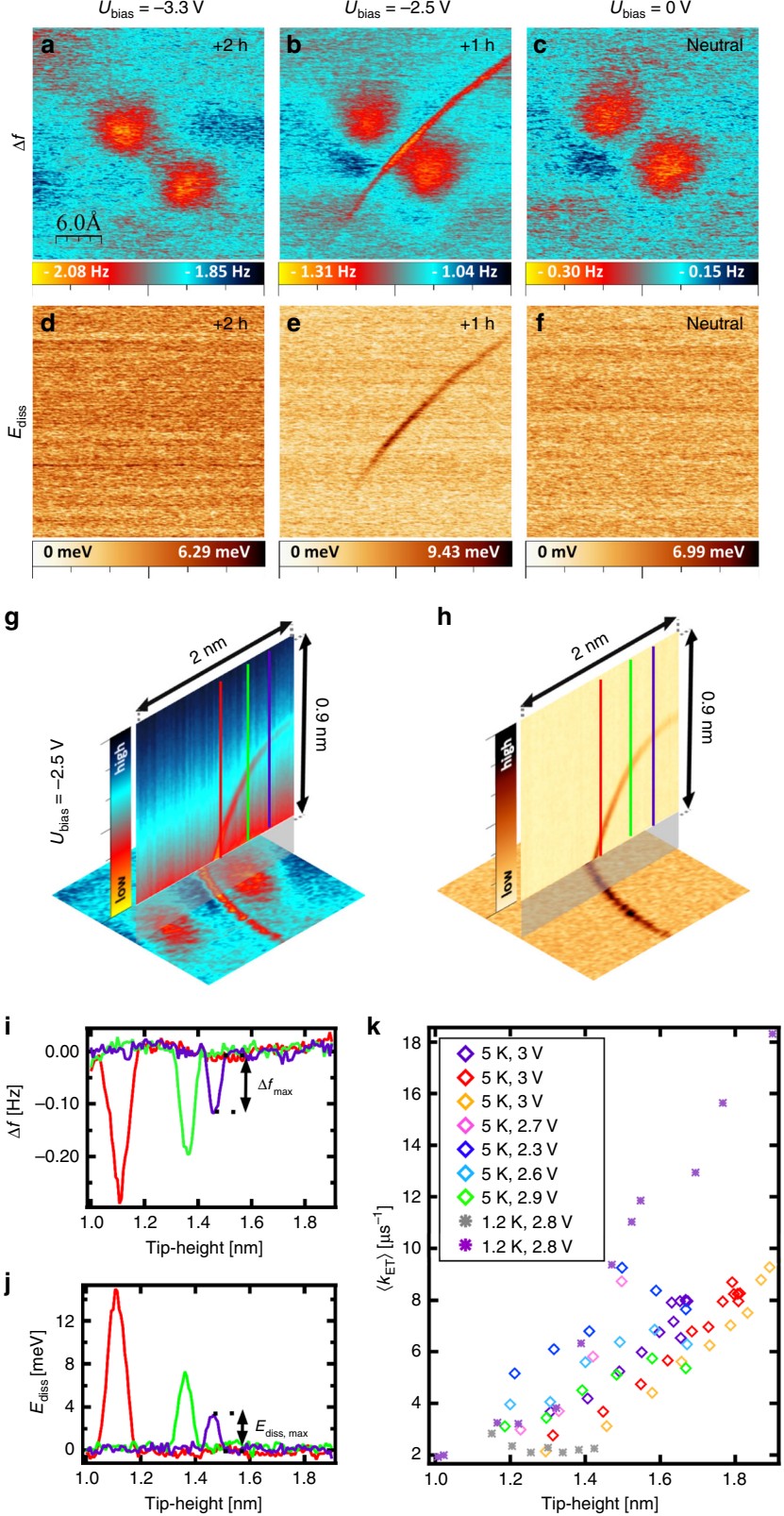

each redox center. This is supported by the distinct alignment of two Fc unit with the substrate (see Supplementary Fig. 1) and pronounced atomic relaxation of the whole system upon the charging processes predicted by DFT calculations, see Supplementary Figs. 7 and 8. Importantly, the ratio between the peak values of the $\Delta f$ and $E_{diss}$ signals makes it possible to determine an effective switching rate $k_{ET}$ related to the phase delay

between the tip oscillation and the electron transfer (ET)[31,34]:

$$k_{ET} = \frac{2\pi^2 kA^2 \Delta f_{max}}{E_{diss,max}}, \quad (1)$$

where $A$ is an oscillation amplitude, $k$ the stiffness of the AFM probe; $\Delta f_{max}$ and $E_{diss,max}$ are the peak values of the frequency shift

**Fig. 2 Experimental detection of the charged state. a–c, d–f** A simultaneously taken series of the constant-height frequency shift and dissipation images taken at the same tip-sample distance at three different bias voltages: $U = -3.3$ V, $U = -2.5$ V and $U = 0$ V (corresponding to double positively charged molecule, once positively charged molecule and neutral molecule). The scanning area is $3 \times 3$ nm$^2$, oscillation amplitude $A = 40$ pm for each image, temperature 5 K. **g, h** Simultaneously taken frequency shift and dissipation $x$-$z$ images of the characteristic sharp ("singularity") line (once positively charged molecule). The vertical cut is $2 \times 0.9$ nm$^2$ with the gap of 0.1 nm between the base image and perpendicular image of the singularity line. The perpendicular image is taken with a bias voltage $U = -2.5$ V. The base image of bisFc is a constant height frequency shift and dissipation image, taken with a bias voltage $U = -2.5$ V (once positively charged molecule). The scanning area is $2 \times 2$ nm$^2$. **i** $\Delta f(z)$ charging curves taken at three different positions, see the red, green, and purple lines marked in **g** and **h**. All curves are shown after background subtraction and tip-height alignment, as described in Supplementary Methods. **j** Dissipation signal $E_{\text{diss}}(z)$ corresponding to $\Delta f(z)$ spectroscopies. Background dissipation was subtracted. **k** Effective transfer rates $k_{\text{ET}}$ obtained from experiments at two different temperatures, various bias voltages and different tips.

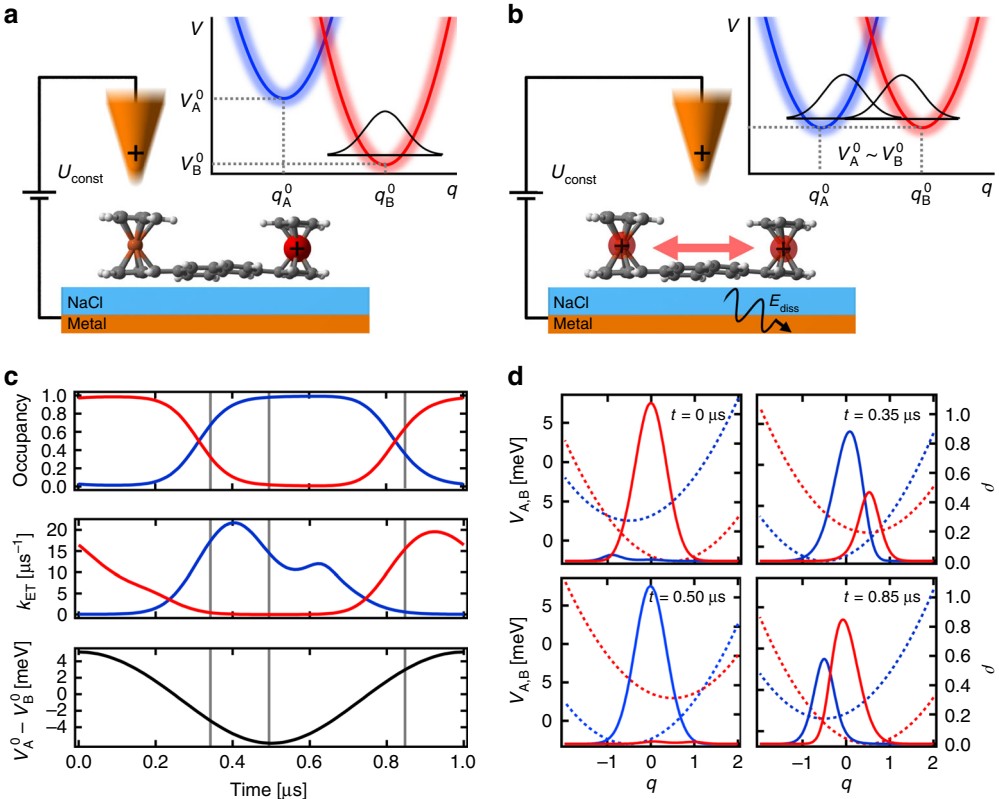

**Fig. 3 Mechanism and dynamics of charge switching within the molecule. a** Schematic representation of once charged bisFc molecule on a NaCl layer. The positively charged tip is located above ferrocene unit A and repels a positively charged hole that occupies the second unit, ferrocene unit B (to minimize its energy). The inset represents the energy levels of both ferrocene centers. **b** A representation of the situation when the tip is in such a position between the ferrocene centers where the energetic levels of both centers are comparable and the charge can move between them: here, the singularity feature shows up. **c** The blue (ferrocene A) and red (ferrocene B) curves in the upper panel represent the occupation probability for the charge on ferrocene A or B during one oscillation cycle of the tip. The middle panel shows the (q-integrated) transfer rate from ferrocene B to A (blue curve) and A to B (red curve). The black cosine curve in the lowest panel represents the energy offset $\Delta E(t)$ between the A and B, which directly relates to the position of the tip during one oscillation cycle. **d** The evolution of the wave packets and energetic levels of both ferrocene centers at four different times: $t = 0$ μs, $t = 0.35$ μs, $t = 0.50$ μs and $t = 0.85$ μs. The dashed lines represent the potential surfaces for bisFc with the charge localized at ferrocene center A (blue) and with the charge at center B (red). The solid lines show the position and amplitude of charge-resolved probability wave packets during one period.

and the dissipation signal, respectively, in the sharp line, as shown in Fig. 2i, j and Supplementary Fig. 9. We carried out a series of experiments to understand the dependence of the effective ET rate $k_{\text{ET}}$ on the tip-sample distance $z$, bias voltage $U$ and temperature $T$, shown in Fig. 2k. We can see that the effective ET rate $k_{\text{ET}}$ is practically independent of temperature $T$ and applied bias voltage $U$ (see also Supplementary Fig. 10), but it slightly increases with the tip retraction from the surface. Note that the $z$-dependence was extracted from a 2D scan in the $xz$-plane, during which the tip meets the position of the resonance feature at different heights (different $z$) for different positions along the molecule (different $x$), see vertical lines in Fig. 2g, h.

The weak dependence on the applied bias $U$ indicates that the electrostatic interaction is mostly driven by the redox state and the permanent (rather than bias-induced) charge on the tip. On the other hand, the negligible effect of temperature on the effective ET rate $k_{\text{ET}}$ suggests that the electron transfer is not thermally activated, violating the canonical detailed balance condition. This observation seemingly contradicts standard rate theories, e.g., Marcus theory[35], where the electron transfer rate $k_{\text{ET}}$ between two redox states depends exponentially on temperature $T$ as $k_{\text{ET}} \approx |M|^2 e^{-\Delta G/k_{\text{B}}T}$. Here $M$ is the electronic coupling between two redox centers, which is determined by the electronic and geometric properties of the molecule, mainly the

molecular linker connecting the redox centers[36,37]; $k_B$ is the Boltzmann constant, and $\Delta G$ means the activation energy of ET process. A possible explanation of this non-standard behavior lies in the fact that the ET process is externally driven by the oscillating probe. External driving does not allow the system to reach its thermal equilibrium. The Gibbs energy is time dependent, and the ET process is observed mostly when its value periodically crosses zero, i.e. the systems is in the so-called Marcus 'inverted' region[35]. Consequently, this may suppress the exponential sensitivity of the switching process to temperature.

**Numerical model of electron transfer driven by AFM probe.** To get more insight into the ET mechanism within the molecule, we carried out numerical simulations of the dynamics of an oscillating AFM probe coupled electrostatically to the ET process between two redox centers, changing their redox states continuously. Due to this coupling, the AFM probe experiences time-varying forces, which define its detected frequency shift and the dissipation energy signal, respectively.

In our model, the charge, which is transferred between the two centers, interacts with an environment formed by the remaining degrees of freedom (DOF) of the molecules and those of the substrate. The exact form of the interaction between the charge and the environment is not known. Like in the usual description of spectroscopic experiments[38], one has to resort to an effective model description of the environment and the system–environment interaction. Without any loss of generality, we can divide the environmental DOF into two kinds of modes depending on their speed of relaxation (reorganization) with respect to the charge transfer between the redox centers. The first DOF reorganize faster than the characteristic charge transfer time while the other DOF reorganize on a comparable or slower time scale. The fast DOF is treated within the established Marcus theory, and they will be characterized by certain reorganization energy $\Lambda$. This energy corresponds to the energy of the electron, which is immediately dissipated into the bath upon the charge transfer between the two redox centers.

The slow DOF, on the other hand, will be simulated by postulating explicit (harmonic) potential energy surfaces (PES) for the situation with the charge on centers A and B, respectively. The probability distribution of the charge states is represented by an effective wave packet of probability density in the space of the explicit DOF (for each of the two charge locations separately). This wave packet is being propagated in the presence of the oscillating AFM probe. We denoted the explicit DOF as slow, but we can in fact vary the speed of the corresponding reorganization process from slow to fast as a parameter of the model. Therefore, we will rather refer to them as explicit modes in the rest of this discussion. Similarly, we will refer to the fast DOF as the Marcus DOF. The importance of the explicit DOF to describe the electron transfer process is demonstrated by better agreement of our full numerical model with experimental evidence, as compared to what we call an all-Marcus model (a model which only includes the Marcus DOF). A detailed discussion of physical interpretation for the two kinds of DOF is provided in Supplementary Note 5.

Each redox center is described by a potential surface $V_{A,B}(q)$, for the charge at centers A and B, respectively, see Fig. 3a, b. Variable $q$ is some effective nuclear coordinate, which tends to relax according to the location of the charge. The equilibrium configuration of the explicit DOF is different for the two possible charge locations; this difference is represented by different values of $q$ for which the respective potentials $V_A(q)$ and $V_B(q)$ reach their minima. The dynamics of the probability wave packets within the potentials is described by coupled differential diffusion (Smoluchowski) equations for the charge densities $\rho(q,t)$ located

on the two redox sites[38] (for details, see Supplementary note 2). The equations are coupled by transfer rates $k_{AB}(q,t)$, $k_{BA}(q,t)$ between the centers. The ET rates depend on the energy difference between the configurations represented by wave packets at the two potential energy surfaces, and they can be described by Marcus theory[39,40], which gives, for transfer of charge from site A to site B:

$$k_{BA}(q,t) = \frac{2\pi}{\hbar}|M|^2 \frac{1}{\sqrt{4\pi\Lambda k_B T}} \exp\left(\frac{-(V_A(q) - V_B(q) + \Delta E(t) - \Lambda)^2}{4\Lambda k_B T}\right)$$

(2)

(and analogously for $k_{AB}$), where $V_{A,B}(q)$ are potential energy surfaces of each redox center, $\Delta E(t)$ represents the time-dependent offset between two potential energy surfaces $V_{A,B}(q)$ due to the variation of the electrostatic field of the probe during the tip oscillation. The reorganization energy $\Lambda$ represents the quantum of energy, which is dissipated by the Marcus DOF during the ET process from the molecule to the environment. This is because the transfer most probably occurs, in accord with Eq. (2), in a resonance condition which is established when the energy difference between the two sites $V_A(q) - V_B(q) + \Delta E(t)$ becomes equal to $\Lambda$. A detailed description of the numerical model and underlying simulations can be found in Supplementary Notes 2 and 4.

Figure 4a, b displays simulated maps of the frequency shift $\Delta f$ and dissipation energy using optimized parameters corresponding

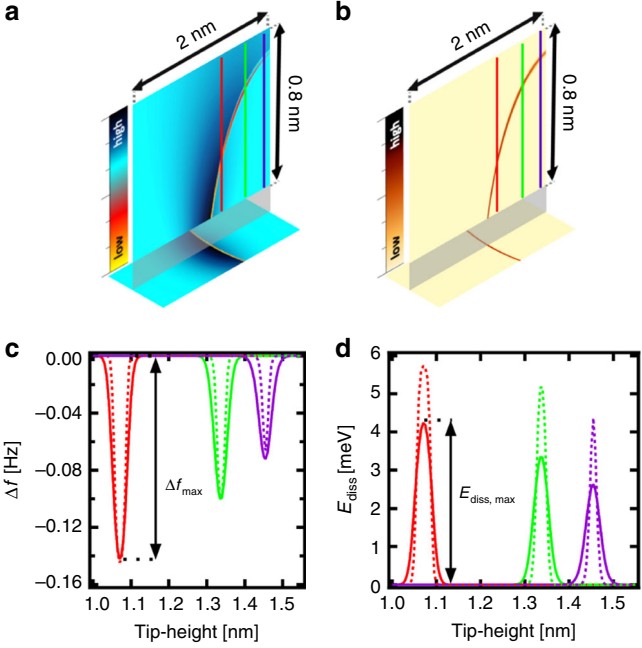

**Fig. 4 Results of the simulations. a** A simulated frequency shift $x$-$z$ image of the singularity line (once positively charged molecule). The vertical cut is $2 \times 0.8\,nm^2$. **b** The simulated dissipation $x$-$z$ image of the singularity line (once positively charged molecule). The vertical cut is $2 \times 0.8\,nm^2$ with a gap of 0.1 nm between the base image and perpendicular image of the singularity line. **c** Calculated $\Delta f(z)$ spectroscopies for three different cuts through the singularity feature corresponding to three experimental $\Delta f(z)$ curves, see Fig. 2i. The solid line is calculated for $T = 5\,K$ and dashed lines correspond to $T = 1.2\,K$. The curves were calculated along the paths highlighted in **a** and **b**. **d** Calculated $E_{diss}(z)$ spectroscopies for three different cuts through the singularity line, along the same paths as in **c**, corresponding to three experimental $E_{diss}(z)$ curves, see Fig. 2j. The solid line is calculated for $T = 5\,K$ and the dashed lines correspond to $T = 1.2\,K$. The amplitude of tip oscillation was set to 40 pm in both cases.

to an intermediate relaxation rate of the explicit DOF, when the wave packet cannot reach the equilibrium between two consecutive charge switches.

The simulation reproduces the character of the singularity line observed in the experiment well, cf. Fig. 2g, h. According to the numerical simulations, the spatial distribution of the singularity line in both the frequency shift and dissipation energy channels is intimately related to the electrostatic field located on the probe (see Supplementary Note 3 and Supplementary Fig. 6), which may change after the tip treatment, as often observed experimentally (Supplementary Fig. 5). The location of the singularity line in space is basically determined by the condition $\Delta E(t) \approx 0$.

**Comparison of the numerical modeling to experimental measurements.** From Fig. 3c, we can see that in our model the values of the rates $k_{AB}$ and $k_{BA}$ vary from 0 to ~20 MHz during one oscillation cycle. The average rate $\langle k_{ET} \rangle$ calculated from the simulated data set using Eq. (1) gives values in the range of 6–8 MHz for different tip-sample distances. These values are in a good agreement with the experimental evidence.

More interestingly, the calculated effective ET rates $k_{ET}$ using Eq. (1) increase only slightly from 1.2 K to 5 K, from 3 to 5 MHz respectively, having the same order of magnitude, far beneath the exponential dependence of the equilibrium Marcus theory transfer rate. Note also that the relation between the effective ET rates $k_{ET}$ given by Eq. (1) and the transfer rates $k_{AB}(q,t)$ and $k_{BA}(q,t)$ given by Eq. (2) is quite complex. The effective rate $k_{ET}$ depends not only on how large the transfer rate is near its peak value but also on the phase of the tip oscillation in which the rate is non-negligible. A phase delay of the rate maxima with respect to tip oscillation is especially sensitive to the reorganization energy $\Lambda$, while the peak magnitude is given by the coefficient in front of the exponential in Eq. (2). The effective rate $k_{ET}$ evaluated according to Eq. (1) is then sensitive to the hopping element $|M|$ as well as to the reorganization energy $\Lambda$ and (to a lesser extent) also to the temperature. More detailed discussion about the relation between rates $k_{ET}$, $k_{AB}(q,t)$ and $k_{BA}(q,t)$ is provided in Supplementary Note 8.

The moderate temperature dependence can be understood by a detailed analysis of the temporal evolution of the $q$-dependent wave packets and the population of the redox states by the charge in the steady-state regime (see Fig. 3 and Supplementary Movie 1). Figure 3c reveals the correlation between the occupancy of the two redox centers, the switching rate $k_{ET}$ and the probe oscillation. The black line shows a cosine function that represents the tip oscillation. The blue and red curves are the probabilities that the respective redox center is occupied by the charge. The charged state completely switches its location between the two redox centers whenever one of the probabilities changes from one to zero. The ET transfer delay can be understood as the average time needed for the ET to be completed after the tip passed through the equilibrium position around which it is oscillating. During the probe oscillation, the potential wells of the redox centers continuously modulate their energetic positions due to the electrostatic interaction of the AFM probe. The instantaneous ET rate $k_{BA}$, given by Eq. (2), is modified accordingly, and it becomes maximal when the condition $V_A(q) - V_B(q) + \Delta E(t) - \Lambda = 0$ is fulfilled during the oscillation cycle. After the ET occurs between the two redox centers, the probability wave packet starts to relax (reorganize) toward the new equilibrium position. Meanwhile, because of the tip motion, the new charge state of the molecule becomes energetically unfavorable, and the ET happens again, in the reverse direction, initiating a relaxation process on the other redox center again. When the tip oscillation occurs on a comparable time scale as the wave packet relaxation

(reorganization of the environment), the wave packets never reach their equilibrium positions before the next switch happens. Figure 3d represents the steady-state dynamics of the wave packets during the oscillation. Importantly, the wave packets are out of equilibrium, located in the near-activationless region close to the potential intersection. Consequently, the temperature dependence of the transfer rate is suppressed, as observed in the experiment. Thus, the key ingredient of the weak temperature dependence of the effective switching rate $k_{ET}$ is the non-equilibrium character of the charge distribution between the redox centers and continuous modulation of the energy levels of redox centers by a periodically oscillating probe.

In our model, both the Marcus and explicit DOF may contribute to the dissipation $E_{diss}$ signal. Our numerical simulations using different speeds of the relaxation process of the explicit DOF reveal that their contribution to the detected dissipation signal in the AFM is not significant unless they reorganize on a time scale comparable to or faster than the tip motion. However, if they reorganize too quickly, they can be added to the fast DOF which we simulate by the Marcus theory. On the other hand, when the explicit DOF reorganize slower than the average rate of the transfer, they do not have the time to relax before the next charge transfer takes place. Consequently, only a negligible fraction of the reorganization energy from the very slow explicit DOF contributes to the dissipation signal detected in AFM measurements.

The simulations reveal that the shape of the $E_{diss}$ (z) peak along z-distance is strongly affected by a character of the ET dynamics. On Supplementary Fig. 11, we compare three different simulations using different description of the ET dynamics: (i) the model with both the explicit and the Marcus DOF (corresponding to Fig. 4c, d), (ii) a model including only the Marcus DOF; and (iii) a simple model where the ET occurs immediately at certain z-distance, where a difference between energies of the two redox centers equals a constant threshold energy $\delta E$. We can see that this simplest model provides a square shape of the $E_{diss}(z)$ peak, while the all-Marcus models show a more rounded shape and the full model a sharp peak similar to the experimental observations, see Fig. 2j. To understand the origin of the distinct shapes, one should look at ET probability per one oscillation cycle along z-distance, shown on Supplementary Fig. 12. We see that the character of the $E_{diss}(z)$ peak is intimately correlated to the character of the ET probability along the z-distance. Thus, the shape of $E_{diss}(z)$ peak provides detailed information about the ET dynamics within a measured system.

Next, we will discuss a comparison between the full model including the explicit DOF and the all-Marcus model. In the case of the full model, the relaxing wave packet manages to dissipate a certain fraction of its potential energy following the charge transfer before another charge transfer takes place again. In principle, we could still approximately include the explicit DOF into an effective Marcus theory description even in this intermediate case, namely by adding an effective contribution of the explicit DOF to the reorganization energy $\Lambda$. The numerical simulations using only the Marcus DOF with larger reorganization energy $\Lambda$ can roughly reproduce the results of the full model with the explicit DOF. But the all-Marcus model provides more rounded $E_{diss}(z)$ peaks than observed in the experiment, see Supplementary Fig. 11. For an unambiguous disentanglement of the relative contribution of the DOF modes with different time scales to the detected dissipation signal in AFM, we would need to measure at different oscillation frequencies of the AFM tip. Unfortunately, this option is not experimentally feasible in the current AFM setups. Another option is to analyze the detected signal as a function of oscillation amplitude. According to the amplitude dependence analysis,

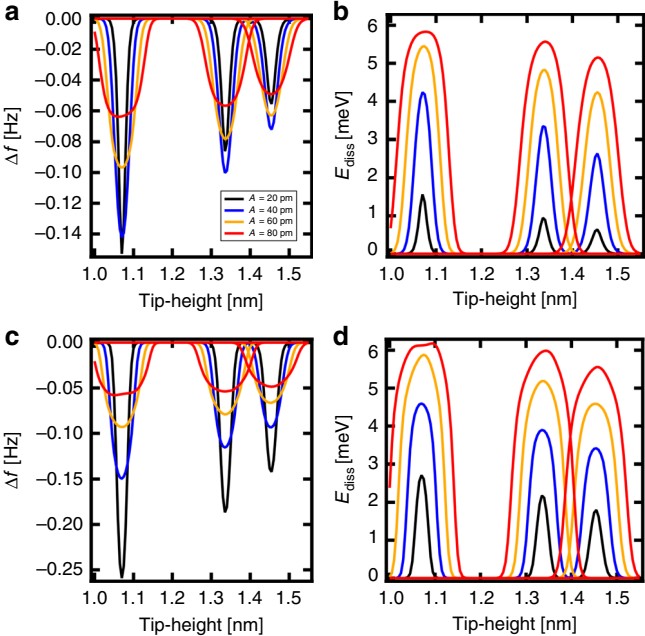

**Fig. 5 Dependence of the simulated frequency shift and dissipation signal on the oscillation amplitude. a** The frequency shift and (**b**) the dissipated energy obtained with the full model with both the Marcus and explicit DOF for different tip-heights and different tip amplitudes ($A = 20$, 40, 60 and 80 pm); (**c**) the frequency shift and (**d**) the dissipated energy obtained with the all-Marcus model (without the explicit DOF) for different tip-heights and different tip amplitudes ($A = 20$, 40, 60, and 80 pm). The simulations are performed for 5 K.

carried out in Supplementary Note 9, see Supplementary Figs. 13 and 14, our full model provides better agreement with the experimental evidence over the all-Marcus model. Namely, the experimental data shown in Supplementary Figs. 13 and 14 reveal non-monotonic dependence of amplitude of the frequency shift on the tip oscillation amplitude. First, the maximum of the frequency shift increases, while for amplitudes larger than $A = 80$ pm it drops again and the peak becomes broader. This non-monotonic behavior cannot be reproduced in the all-Marcus model, where the frequency shift monotonically decreases with larger amplitude, see Fig. 5c. The evolution of the dissipation signal with the oscillation amplitude is also better captured by the full model.

Finally, we would also like to note that the experimental data shown in Fig. 2k reveal a slight enhancement of the effective ET rate $k_{ET}$ with increasing tip-sample distance. This trend is not reproduced in our model. This can be related to the fact that, in our model, we consider the electronic coupling $M$ between two redox centers to be constant, independent of the tip-sample distance. But apparently the electronic coupling can be affected by the proximity of the electric field to the probe.

## Discussion

Most processes in nature occur out of the equilibrium (e.g. solar cells excited by light are just such an example). Therefore, it is not surprising that both experimental and theoretical efforts to study systems out of the equilibrium have been increasing recently. In this manuscript, we demonstrated that AFM can control a mixed-valence state in a single molecule with multiple redox centers. We also showed that AFM can probe the quantum dissipation processes related to the electron transfer within the molecule driven out of the equilibrium. To our knowledge, this is the first time that the out-of-equilibrium ET processes have been studied within a single

molecule. We believe that this work opens a new route to study the quantum dissipation processes on a single-molecular level addressing fundamental questions, e.g. how fast the ET processes are, what is the role of the substrate, does the electronic correlation play a role, or how functional groups and thermal structural fluctuations affect the ET process. Along with the recent advances in on-surface UHV chemistry, we can consider the possibilities to study the electron transfer mechanisms and underlying quantum dissipation processes in the presence of multi-electron charge states in well-defined molecular assemblies. These experiments will address fundamental questions of mutual interaction and dynamics of multiple charges, and the possible formation of condensed states. Finally, concerning the theoretical description, other formalisms designed to describe the dynamics of the open quantum systems such as Caldeira-Leggett[1–3] could be applied to the description of the experiment as well (as discussed in Supplementary Note 6). We believe that the presented experimental setting and the controlled character of the experiments it enables provides an ideal experimental material for refinement and development of the methods of open quantum systems theory devoted to the problem of electron transfer in molecular systems and nano-structures.

## Methods

**ncAFM experiment details.** The microscopy measurements were performed with a low temperature JT-SPM, manufactured by SPECS Surface Nano Analysis GmbH, in UHV conditions in the range of $10^{-10}$ mbar and at a temperature range between 1.2 K and 5 K. The imaging of the molecules was performed using a Kolibri sensor (also called needle-sensor)[41], with a chemically etched tungsten tip. This setup allows for simultaneous measurement of the frequency shift and tunneling current between an oscillating tip and the sample surface. The measurements were performed using different Kolibri sensors, each of them having slightly different resonance frequency $f_r \sim 1$ MHz. We considered the stiffness $k = 1080$ kN m$^{-1}$ for all sensors. The quality factor $Q$ of Kolibri sensors was determined independently for each session, so that its variation with tip modification or after a sensor exchange is taken into account. Upon insertion into UHV, the sensor tip was cleaned by Ar$^+$ sputtering.

We used mono-crystals of Ag(111). The sample was degassed in UHV at 550 °C for 6 h and subsequently cleaned by cycles of Ar$^+$ ion sputtering and annealing to 550 °C.

The cleanliness of the surfaces was confirmed using low energy electron diffraction (LEED).

The NaCl/Ag(111) surface was prepared by evaporating NaCl at the temperature 600 °C on clean Ag(111) crystal held at room temperature. The sample was shortly post annealed after the evaporation. The sample was directly transferred to the scanning head and cooled down to 5 K after preparation of NaCl film.

BisFc molecules were thermally deposited from a tantalum crucible directly on the sample held at 4.2 K. Temperature of the bisFc evaporation was 185 °C. The pressure in the chamber did not exceed $9 \times 10^{-10}$ mbar. The temperature in the microscope head reached 8 K at the end of evaporation. For details on synthetization of bisFc molecules see Supplementary Figs. 15 and 16.

**Background subtraction and peak parametrization in $\Delta f(z)$ spectra and $E_{diss}(z)$ spectra.** To compare the experimental values of the dissipation signal $E_{diss}$, we converted the signal, originally measured as the excitation voltage needed to keep constant oscillation amplitude, into energy dissipated per one oscillation cycle. The dissipation signal was (in contrast to the frequency shift) mostly featureless apart from the sharp line which we attribute to the electron transfer between the two redox centers. The flat featureless background signal, which corresponds to a background dissipated energy solely caused by the finite $Q$-factor, was evaluated by formula:

$$E_{diss}^o = \pi k A^2 / Q, \tag{3}$$

where $k$ is the cantilever stiffness, $A$ is the oscillation amplitude. By comparing the value of background excitation voltage to the dissipated energy $E_{diss}^o$ calculated with the above formula, we obtain the desired conversion factor for the dissipation signal.

In order to analyze more quantitatively the sharp feature that we see in AFM images when the charge continuously swaps between the molecular redox centers, we measured sets of $\Delta f(z)$ and $E_{diss}(z)$ scans perpendicular to the surface as shown on Fig. 2g, h. Wherever the z-scan crosses the charge-swapping feature, a peak appears in the dissipation spectrum and a similar negative peak (dip) appears in the frequency shift. To estimate the effective transfer rate $\langle k_{ET} \rangle$ using Eq. (1), we need to estimate the maximum amplitude of the frequency shift $\Delta f_{max}$ and the dissipated energy $E_{diss,max}$. The estimation of $E_{diss,max}$ is trivial due to the flat character of the

background dissipation signal, see Fig. 2h. The background superimposed on the corresponding dips in the frequency-shift channel is however more complex. We fitted the latter background with a rational function of the form

$$\Delta f_{backg.}(z) = \Delta f_0 - C_0/(z - z_{back})^2, \qquad (4)$$

with three adjustable parameters, $\Delta f_0$, $C_0$ and $z_{back}$. We note that the choice of the power of $-2$ is not unique, picking $-3$ instead of $-2$, for example, would work just as well for most spectra.

One component of the background also arises due to the electrostatic interaction of the tip with the extra charge of the molecule. Note that the peaks in the spectra form because of a periodic switching of charge location synchronized with tip oscillation. The force that corresponds to time-averaged location of the charge should be nevertheless still counted as a contribution to the background. This particular background component is expected to form a sharp cusp right at the position of the peaks, as it is at this point where the location of the charge changes abruptly. Surprisingly, this background component is weak enough compared to the rest of the background so as not to spoil the regression that uses the smooth rational function specified above. In contrast, the background that arises because of interaction with the time-averaged charge distribution is going be the only one we will need to subtract from the simulated spectra, as we explain further in Supplementary Note 7.

Simultaneously with the background fitting, we fitted a Gaussian function to the frequency-shift dip which is for us the relevant useful signal. So in the end we used the following analytical function to be fitted onto the $\Delta f(z)$-curves from experimental data:

$$\Delta f(z) = -\Delta f_{max} \exp\left(-\frac{(z - z_{dip})^2}{2\sigma^2}\right) + \Delta f_0 - \frac{C_0}{(z - z_{back})^2}, \qquad (5)$$

The above function contains six free parameters altogether, three of them to reproduce the background as already discussed above and the other three to describe the dip, namely its height $\Delta f_{max}$, position $z_{dip}$ and width $\sigma$. Similarly, the peaks in dissipated energy were fitted with a curve characterized by four parameters, one for a constant background and the other three again for a Gaussian-shaped peak.

We note here that the precision of our measurement did not allow to distinguish a Gaussian shape of the peaks from, say, a Lorentzian one. Both Gaussian and Cauchy-Lorentz functions fitted the peaks equally well within the tolerance given by the measurement noise. Indeed, the numerical model we develop to explain the measured data does not predict either exactly Gaussian or Lorentzian shapes, but it typically produces some more general bell-shaped peaks that can be approximated e.g. by the Gaussian functions reasonably well.

In order to compare experimental and simulated $\Delta f(z)$ and $E_{diss}(z)$ spectra we need to align the tip-sample distances. The absolute value of the tip-sample distance between different experiments is unknown. Therefore, we decided to align the experimental data obtained from different sessions in the way that the lowest end of the range of each data set coincides at the same tip-sample height. The tip-sample distance has been estimated from theoretical model, which is defined as the distance between the outermost effective static charge on tip and charge on ferrocene units.

**DFT calculations**. We carried out DFT calculations of neutral, $+1\,h$ and $+2\,h$ charged molecules on NaCl substrate using the FHI-aims package[42], which is an all-electron full-potential DFT software based on numeric atom-centered basis functions. All geometry optimizations and electronic structure analyses have been performed using the B3LYP hybrid exchange-correlation functional[43] as implemented in FHI-aims[44]. Van der Waals interaction was described by the Tkatchenko-Scheffler method[45]. The hybrid functional was mandatory to describe correctly the localization of single-electron charge states on a single Fc unit. The surface was modeled by a three-atomic-layers thick slab of NaCl(100) with periodic boundary conditions (PBC). One super cell comprised $(6 \times 3)$ surface unit cells of NaCl and was 2.5 nm thick in the direction perpendicular to the surface, including the NaCl slab, the adsorbed molecule and a vacuum gap. Only $\Gamma$ $k$-point was used to sample the Brillouin zone. The two subsurface layers of NaCl were fixed in their bulk positions. The remaining atomic positions, that is those in the molecule and the topmost NaCl layer, were allowed to relax until the remaining atomic forces shrunk below $10^{-2}$ eV Å$^{-1}$. The charged systems were calculated as spin polarized.

## Data availability
All data are available upon request from the authors.

## Code availability
Simulation package of charge dynamics is available here: https://github.com/mondracek/DissipativeChargeTransferInAFM.git

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

## Acknowledgements

Authors acknowledges fruitful discussions with Tomáš Novotný and Artem Ryabov. This work was supported by the Czech Science Foundation (Reg. No. 18-18022 S, 18-09914 S), MEYS LM2015087, Operational Programme Research, Development and Education financed by European Structural and Investment Funds and the Czech Ministry of Education, Youth and Sports (Project No. SOLID21-CZ.02.1.01/0.0/0.0/16_019/0000760) and the Institute of Organic Chemistry and Biochemistry, Academy of Sciences of the Czech Republic (RVO: 61388963). P.J. acknowledges support from Praemium Academie of the CAS. Computational resources were provided by the CESNET LM2015042 and the CERIT Scientific Cloud LM2015085, provided under the programme "Projects of Large Research, Development, and Innovations Infrastructures".

## Author contributions

P.J. and I.S. conceived the project and designed the experiments. J.B., O.S., and M.S. performed and analyzed the SPM experiments. M.O., P.M, T.M., and P.J. designed the theoretical model and performed simulations and their analysis. P.H., J.R., I.G.S., and I.S designed and synthesized molecules. P.J., M.O., T.M., P.M., and I.S. co-wrote the paper. All authors discussed the results and commented on the manuscript.

## Competing interests

The authors declare no competing interests.
