## [Peer Review File · Nature Communications]

Reviewers' comments:

Reviewer #1 (Remarks to the Author):

I do not recommend this paper for publication in Nature Communication.

Comments:

The manuscript by Berger et al. deals with intramolecular electron transfer between two ferrocene redox centers on a NaCl insulating layer with NC-AFM. The authors demonstrate the reversible charging of the molecule with 1 or 2 holes. Afterwards, they evaluate the electron transfer rate between ferrocene centers in the 1 hole charged state by the measurement of the dissipation and frequency shift at different temperatures and bias voltages.

From the data, the authors conclude in a weak dependence of the transfer rate on the applied bias voltage and temperature.

To explain the results, the authors develop a numerical model based on the interaction between a charged tip and a molecule.

Results are of great interest and most of the methodology is given in supplementary materials, however:

1) This article is really long, and the reader must refer to the supplementary materials to understand most of the presented results.

2) There are a lot of typographic errors in the figure numbers, and some important data are missing, for example the fig. S3 with $df(U)$.

3) On Fig. 2 (k), the authors plot the average electron transfer rate ($\langle K_{eff} \rangle$) as a function of the tip height and sample temperature, for different tips.

This figure is too small, the authors should improve it.

Moreover, from Fig.2 (l) and (j), for a given bias, the resonance of the tip with the intramolecular transfer rate only occurs at a given distance from the surface. As a consequence, for one bias, $df(z)$ only give one evaluation of $\langle K_{eff} \rangle$ at a given distance and there is no experimental possibility to change the height at which the resonance appears.

Could the authors give some explanations on the calculation of $\langle K_{eff} \rangle$ for others heights observed experimentally and presented in fig. 2(k)?

This point is fundamental, since it is the justification for the low bias dependency and increasing transfer rate with height.

For these reasons, I do not recommend this paper for publication in Nature Communication.

Reviewer #2 (Remarks to the Author):

The manuscript reports an atomic force microscopy (AFM) study of single molecules containing two redox centers adsorbed on a thick layer of NaCl. Depending on the applied voltage to the tip the authors find the molecules in different charge states having 0, 1, or 2 holes with one hole located at each redox center (here ferrocene molecules). Under certain conditions, tip voltage and position, they find that the singly charged molecule exhibits electron transfer between the two ferrocene units within an oscillation cycle of the AFM tip for a single molecule. Theoretical modeling allows the authors to simulate the experimentally observed frequency shifts and energy dissipation and provide insight into the energy relaxation behavior of that molecule.

The manuscript is well-written, the presented data is of high quality, and the theoretical model appears valid and reproduces well the experimental findings. The results are generally interesting and hint at the possibilities to study redox reactions at the single molecule level with insights into the energy dissipation mechanisms. However, in my view it is not clear in as much the observations and conclusions are addressing prominent questions in this field. Since the charge transfer between the redox centers is driven by the AFM tip and its electric field, it remains open to me what are the intrinsic properties of the molecules how this can be used to study electron transfer processes between molecules and surfaces or intramolecular charge rearrangements independently of the measurement scheme. The system is driven artificially out-of-equilibrium by the tip, where would one find such a situation? It appears that the investigation of this molecule and the findings are isolated and a particular case, however, well studied. How does this connect to other relevant investigations in this field of molecule redox properties other than that electron transfers can be driven by an AFM? For instance, what would be the electron transfer rate for the single charged molecule in the absence of the AFM tip? Can one extend this to charge transfer processes between molecules and surfaces and study those dissipation processes which is relevant for dye-sensitized solar cells or functionalized surfaces for catalysis? If the authors can provide further information on this issue and extend their conclusions beyond an isolated case study, I would find the manuscript suitable for publication in Nature Communications.

Besides that I have a couple of questions and suggestions to be considered in the revision.

The „Quantum“ in the title is not clear to me and the transfer rates, although worked out for an electron density distribution, appear classical to me. Is there any phase coherence in the process? Is the dissipation energy quantized, i.e. quantized levels in the local parabolic potential? Can you provide a link to the quantum character of the dissipation in the manuscript text?

The order of the references to the figures must be checked. Figure 4 comes before Figure 3 and also the individual panels are not cited in order.

I appreciate the lengthy presentation and discussion of the theoretical model as it is a central part of the manuscript, however I would find it very useful if the current Figure 4 can be extended to illustrate also the situation of the potential landscape within a single oscillation cycle. The information is essential contained in Figure 4c and d but could be extended by an additional illustration showing the situation over time. I assume that in the insets of Figure 4a and b the energy landscape of the hole is already inverted. This can sometimes confusing when talking about electron transfer but consider actually a hole. In Figure 4b, where is the dissipation coming from when V_a and V_b are equal? Can the parameter Λ be added there? Does caption 4c refer to 4b?

One can imagine that for a symmetric tip there should be a situation when an oscillation cycle of the tip does not change $V_A=V_B$ but only the barrier between the two valleys, i.e. a vanishing $\Delta E(t)$. Has this been observed? The hole should be delocalized over the two centers. „Left“ and „right“ of this position the $\Delta E(t)$ should see a phase change of 180deg. Can the data provide information on the direction of charge transfer with respect to the phase of the oscillation?

The theory considers wave-packets, but would a simpler state theory suffice, i.e. just considering if the hole would be either left or right? In as much tunneling between the two states is included in the theory?

The last page contains a lengthy discussion of the model, however, referring to a figure in the SOM. If this discussion is central to the understanding the corresponding figure should appear in the main text.

Reviewer #3 (Remarks to the Author):

The authors have carried out an extensive study of single electron transfer within a single molecule on an insulating substrate by means of frequency modulated atomic force microscopy under ultra-high vacuum conditions. The molecule aptly chosen consists of two ferrocene redox centers separated by a naphthalene linker. This allows the authors to maintain different oxidation states of

Fe in the two centers through selectively extracting 2, 1 or 0 electrons with appropriate gating of the AFM tip. Very interesting results presented in Figure 2 (authors should correct the numbering in the caption of this figure) show a strong signal that appears as a sharp line in both the measured frequency shift and energy dissipated channels in between the two redox centers in the mixed valence state. This signal is missing in the case of the neutral and the fully oxidized (+2h) charge states. The authors attribute the origin of the effect to a continuous switching of the charge between the two redox centers at a certain tip-sample position with respect to the redox centers. Importantly, the effect shows very little dependence on the tip temperature.

The rest of the paper is on analysis of these two signals using insights from density functional theory-based calculations and Marcus rate equation.

The observation of single electron transfer is not new, although it is nice to see another example of it. The controlling of the mixed-valent state of the molecule with appropriate gating of the AFM tip is a notable contribution. Perhaps the most important contribution of the work is the temperature independence (almost) of the switching rate which the authors interpret as quantum dissipation. While the authors have carried out detailed analysis of the rate using Marcus theory of classical dissipation and effect of the environment, there is room for more fundamental theoretical work. The authors claim, as a result of extensive analysis, that “the weak temperature dependence of the switching rate is solely due to the non-equilibrium character of the electron transfer”, is reasonable and should stimulate further work using quantum mechanical methods for examination of systems that are far from equilibrium. For example, is this an ultrafast process? Do electron correlations play a role?

Authors should proof read the manuscript so careless errors are avoided: labeling of Figure 2, details of Ref. 2, etc.

I recommend publication as the work and its interpretation is novel. It should lead to debate and discussion in the field and trigger more experimental and theoretical work.

Dear Editor,

We are grateful to all three referees for their time spent on the evaluation of our manuscript and for their constructive and valuable feedback. We hope that they will find our argumentation convincing and the new version of the manuscript suitable for publication. All changes made in the manuscript following the referee's comments are marked in red.

Reviewers' comments:

Reviewer #1 (Remarks to the Author):

The manuscript by Berger et al. deals with intramolecular electron transfer between two ferrocene redox centers on a NaCl insulating layer with NC-AFM. The authors demonstrate the reversible charging of the molecule with 1 or 2 holes. Afterwards, they evaluate the electron transfer rate between ferrocene centers in the 1 hole charged state by the measurement of the dissipation and frequency shift at different temperatures and bias voltages.

From the data, the authors conclude in a weak dependence of the transfer rate on the applied bias voltage and temperature.

To explain the results, the authors develop a numerical model based on the interaction between a charged tip and a molecule.

Results are of great interest and most of the methodology is given in supplementary materials, however:

We thank the referee for a positive evaluation of the manuscript and relevant comments, which improved the content of the manuscript. We reply to all the comments one-by-one below.

1) This article is really long, and the reader must refer to the supplementary materials to understand most of the presented results.

We do understand the referee's comment, but we do not see a clear way how to overcome this problem, having in mind that the work contains relatively novel aspects in both experimental and theoretical analysis. For example, the employed numerical model going beyond the standard Marcus model needs to be introduced together with a discussion of relevant parameters. We also try to provide as complete information as possible, so that other groups may potentially reproduce our experimental and theoretical analysis. We therefore believe that the length of our manuscript is justified. We considered trimming the manuscript, but decided against it after finding all its present content significant for the overall presentation of our results.

Note: In order to partially alleviate the problem of many references from the article to the Supplementary material, we have moved Fig. S14 to the article as Fig. 5; see our reply to Reviewer #2's comments below.

2) There are a lot of typographic errors in the figure numbers, and some important data are missing, for example the fig. S3 with $df(U)$.

Thank you for this comment. We revised manuscript accordingly and improved the labelling and ordering of Figures in both the manuscript and the Supplementary information. We added missing Figure S3 into Supplementary information text. The numbering of the figures that follow was changed accordingly (both in the main text and the Supplementary information).

3) On Fig. 2 (k), the authors plot the average electron transfer rate ($\langle K_{eff} \rangle$) as a function of the tip height and sample temperature, for different tips. This figure is too small, the authors should improve it.

We agree with the referee that Figure 2(k) should be rescaled. We changed the arrangement of the panels in Fig. 2, so that now the panel Fig. 2(k) is larger.

Moreover, from Fig.2 (l) and (j), for a given bias, the resonance of the tip with the intramolecular transfer rate only occurs at a given distance from the surface. As a consequence, for one bias, $df(z)$ only give one evaluation of $\langle K_{eff} \rangle$ at a given distance and there is no experimental possibility to change the height at which the resonance appears. Could the authors give some explanations on the calculation of $\langle K_{eff} \rangle$ for others heights observed experimentally and presented in fig. 2(k)? This point is fundamental, since it is the justification for the low bias dependency and increasing transfer rate with height.

We thank referee for this comment. Indeed, we can evaluate the average transfer rate ($\langle k_{ET} \rangle$ or $\langle K_{eff} \rangle$) for different tip-sample distances. As shown in Figure 2g,h, a scan of the xz plane reveals the resonance as a kind of arc shape. Once the tip is allowed to scan, say, along x-direction between two redox centers, the resonance occurs at different tip-height for different x in general. For this reason, it was possible to collect data for the plot shown in Fig. 2h, including the z-dependence, and to apply Eq. (1) of our paper to get $\langle k_{ET} \rangle$. The shape of the arc that corresponds to the spatial location of the resonance differs from molecule to molecule (presumably because of differences in adsorption positions and sometimes also because of a change of the tip), so the range of tip heights for which we were able to acquire the data is not the same across different data sets, but the point is that we were never constrained just to one particular tip height. It could be justly argued that we cannot separate dependence on z from a possible dependence of x. However, we preferred to interpret the results in terms of z dependence as there is no obvious preferred direction along the x axis: Moving along x typically means going closer to one of the redox centers of the molecule while moving farther away from the other one.

Reviewer #2 (Remarks to the Author):

The manuscript reports an atomic force microscopy (AFM) study of single molecules containing two redox centers adsorbed on a thick layer of NaCl. Depending on the applied voltage to the tip the authors find the molecules in different charge states having 0, 1, or 2 holes with one hole located at each redox center (here ferrocene molecules). Under certain conditions, tip voltage and position, they find that the singly charged molecule exhibits electron transfer between the two ferrocene units within an oscillation cycle of the AFM tip for a single molecule. Theoretical modeling allows the authors to simulate the experimentally observed frequency shifts and energy dissipation and provide insight into the energy relaxation behavior of that molecule.

The manuscript is well-written, the presented data is of high quality, and the theoretical model appears valid and reproduces well the experimental findings. The results are generally interesting and hint at the possibilities to study redox reactions at the single molecule level with insights into the energy dissipation mechanisms. However, in my view it is not clear in as much the observations and conclusions are addressing prominent questions in this field. Since the charge transfer between the redox centers is driven by the AFM tip and its electric field, it remains open to me what are the intrinsic properties of the molecules how this can be used to study electron transfer processes between molecules and surfaces or intramolecular

charge rearrangements independently of the measurement scheme. The system is driven artificially out-of-equilibrium by the tip, where would one find such a situation? It appears that the investigation of this molecule and the findings are isolated and a particular case, however, well studied. How does this connect to other relevant investigations in this field of molecule redox properties other than that electron transfers can be driven by an AFM? For instance, what would be the electron transfer rate for the single charged molecule in the absence of the AFM tip? Can one extend this to charge transfer processes between molecules and surfaces and study those dissipation processes which is relevant for dye-sensitized solar cells or functionalized surfaces for catalysis? If the authors can provide further information on this issue and extend their conclusions beyond an isolated case study, I would find the manuscript suitable for publication in Nature Communications.

We thank the referee for his/her time spent on the evaluation of the manuscript and constructive feedback, which helps us to improve the scientific content and readability of the manuscript. Above, the referee expressed his/her concern about an impact of the present study to our understanding of charge transfer processes. We believe that the present work may provide understanding of energy transfer processes in several new aspects.

First, as we mentioned in the introduction, most of electron transfer (ET) and related dissipation energy studies were carried on an ensemble level. However, scanning probe microscopy allows the possibility to study these processes on a single molecule. For example, this may allow better understanding of impact of local environment on the ET processes. The experience from other fields, such as optical spectroscopy, suggests that comparing ensemble and single molecular techniques always opens new pathways to understanding molecular processes, even if those are not studied under equivalent conditions.

Second, as mentioned by the referee, the system is driven out-of-equilibrium. Nowadays, most theoretical descriptions of ET assume of the equilibrium condition, whereby all difficulties associated with introducing the description of non-equilibrium states can be avoided. However, most processes in nature occur out of the equilibrium (solar cells excited by light are just such an example). Therefore, it is not surprising that both experimental and theoretical efforts to study systems out of the equilibrium have been increasing recently.

Third, the Referee #3 wrote: *“While the authors have carried out detailed analysis of the rate using Marcus theory of classical dissipation and effect of the environment, there is room for more fundamental theoretical work. The authors claim, as a result of extensive analysis, that “the weak temperature dependence of the switching rate is solely due to the non-equilibrium character of the electron transfer”, is reasonable and should stimulate further work using quantum mechanical methods for examination of systems that are far from equilibrium. For example, is this an ultrafast process? Do electron correlations play a role? “.* We fully agree with this statement and we hope that this work will inspire other groups to analyze the system with more advanced theoretical tools, e.g. variants of spin-boson model in near future.

Fourth, we anticipate that having control of the formation of single electron charge states, we may attempt to study molecular systems with many redox centers and investigate their behavior under formation of multi-charge states. These experiments will address fundamental questions of mutual interaction, their dynamics and possible formation of condensed states.

We extended the discussion in the conclusions of the manuscript accordingly.

As for the Referee's question about electron transfer rate in the absence of the tip, we indeed observe that the tip does influence slightly the transfer rate. This is evident from height dependence of the transfer rate (see Fig. 2k in the manuscript), where the rate increases with tip-sample distance, but keeping the same order of magnitude ($1/\mu\text{s}$). It seems reasonable to assume that the rate in the absence of the tip corresponds to a far tip-sample

distance limit we found in our experiments, i.e. being of order of ~ 10 $1/\mu\text{s}$.

The „Quantum“ in the title is not clear to me and the transfer rates, although worked out for an electron density distribution, appear classical to me. Is there any phase coherence in the process? Is the dissipation energy quantized, i.e. quantized levels in the local parabolic potential? Can you provide a link to the quantum character of the dissipation in the manuscript text?

By using the term quantum dissipation, we refer to the fact that the mechanism of irreversible loss of energy, which we observe in the experiment, is driven by quantum mechanics. Whether the dissipation is quantum or not cannot be decided based on discussing coherence – there is also classical coherence (William H. Miller, JCP 136 (2012) 210901). Markus theory of electron transfer contains tunnelling term and treats molecular environment quantum mechanically. It is formally equivalent to the Förster resonance energy transfer theory of molecular excitons. Within the latter theory one can show that classical systems do not thermalize the same way as the quantum ones do – classical behaviour is a high temperature limit of the quantum one. The fact that an electronic system relaxes between discrete quantum states towards canonical thermal equilibrium with finite temperature is enabled by the quantum properties of the environment (M. Reppert and P. Brumer, JCP 149 (2018) 234102). Quantum and classical behaviour are classified in this case by direct comparison of quantum mechanical and classical treatments. We show that non-equilibrium quantum treatment satisfies canonical detailed balance condition for the rates while showing a decrease in temperature dependence due to its non-equilibrium nature. Therefore, we believe that the term quantum dissipation is well justified.

The order of the references to the figures must be checked. Figure 4 comes before Figure 3 and also the individual panels are not cited in order.

We agree that the ordering and of the Figures 3 and 4 could be confusing and referencing Figures in text was not done properly. We changed the order of Figures. The figure that was Figure 4 in the first draft is now Figure 3 and vice versa. Also referencing to the figures in the text was changed accordingly. We also changed citations for the Figures S16 and S17 that are referring to the synthesis of 2,6-Diferrocenylnaphthalene.

I appreciate the lengthy presentation and discussion of the theoretical model as it is a central part of the manuscript, however I would find it very useful if the current Figure 4 can be extended to illustrate also the situation of the potential landscape within a single oscillation cycle. The information is essential contained in Figure 4c and d but could be extended by an additional illustration showing the situation over time. I assume that in the insets of Figure 4a and b the energy landscape of the hole is already inverted. This can sometimes confusing when talking about electron transfer but consider actually a hole. In Figure 4b, where is the dissipation coming from when V_a and V_B are equal? Can the parameter Λ be added there? Does caption 4c refer to 4b?

We are pleased that the referee acknowledges the importance of our theoretical model for the discussion in our manuscript. We in fact believe that, given the space limitations, our Fig. 3 (in the new version of the manuscript, previously Fig. 4) already presents the temporal evolution of the model system within one oscillation cycle in quite some detail. As for the potential landscape, the insets of Fig. 3a and 3b show only snapshots at some particular time (say, when the oscillating tip passes its equilibrium position). The whole dynamic of the potential landscape can be best seen in the Movie S1, which we also attach as a supplementary material. In particular, Fig. 3b shows V_A and V_B to be equal, but they are exactly so only in one particular instance in time. The oscillating tip periodically disturbs this equality, as we also

discuss in our manuscript, especially in the paragraph that immediately follows after our first reference to Fig. 3b in the text. What we wanted to convey by Fig. 3b was that the oscillating tip can be placed in such a way that the potential landscape oscillates near the symmetric situation. Then, the minimum of the potential surface V_A becomes the global minimum for about half of the oscillation cycle while the minimum of potential V_B takes over as the global minimum in the other half of the cycle. It is just close to such arrangement where periodic charge transfer accompanied with energy dissipation becomes possible. In the situation depicted on Fig. 3a, in contrast, the global minimum remains on the V_B surface all the time (albeit the depth of the minimum may vary with tip oscillation), so the system eventually settles in this minimum and no charge transfer occurs, hence no dissipation.

The panels of Fig. 3c and 3d in fact already show the temporal evolution of various quantities over one oscillation cycle. In the case of Fig. 3c, the horizontal axis of the plots represents time and spans exactly one period of the oscillation. In Fig. 3d, the four sub-panels show snapshots taken at different times, which correspond to four different representative phases of the oscillation cycle, thus also illustrating the evolution (of the probability distribution and of the potential landscape) within the cycle.

Concerning the distinction between an electron and a hole, the quantities we are showing, such as the potentials, occupation probabilities, etc., apply to a hole. Given the simplicity of our model, however, switching from a hole-centered to electron-centered view would not make much difference. We sometimes allow ourselves to refer to the charge transfer process as an “electron transfer” because of the possible dual view: transferring the hole from one site to the other corresponds to a transfer of an electron in the opposite direction.

We prefer not to include parameter Λ in Fig. 3b. The issue of the role of this parameter may be somewhat confusing and we believe incorporating this parameter in Fig. 3b would only add to the confusion. In particular, we would like to stress that the parameter we denote as Λ is completely independent from the parameters which define V_A and V_B . The potentials V_A and V_B define the effective potential landscape for what we call the “explicit” degrees of freedom in our manuscript, while Λ is an effective reorganization energy for the other degrees of freedom, those which we refer to as “fast” or “Marcus” in the manuscript. Since Λ applies to a different (in a sense complementary) set of degrees of freedom than the potentials, we consider it better not to include both in one figure. We do not introduce any special notation for the reorganization energy of the explicit degrees of freedom in the main text of our manuscript. We only do so in the Supplementary text, where we denote it as E_Λ . We could put this symbol to Fig. 3b in order to indicate how this reorganization energy of the explicit degrees of freedom relates to the depicted potential landscape, but we still prefer not to do so, because we do not refer to E_Λ anywhere else in the main text.

In summary, we believe that the suggested enhancement of the figure that shows evolution of the system over time (Fig. 3 in the newly submitted manuscript) is already covered by the movie submitted as supporting information. Also, we find the representation of the quantity Λ in one image with V_A , V_B to be counterproductive for image clarity. We therefore refrain from making any changes in the figures (except from a small correction in Fig. 3, which stems from a confusion between sites A and B we have spotted in our original figure).

One can imagine that for a symmetric tip there should be a situation when an oscillation cycle of the tip does not change $V_A=V_B$ but only the barrier between the two valleys, i.e. a vanishing $\Delta E(t)$. Has this been observed? The hole should be delocalized over the two centers. „Left“ and „right“ of this position the $\Delta E(t)$ should see a phase change of 180deg. Can the data provide information on the direction of charge transfer with respect to the phase of the oscillation?

If the oscillating tip does not change the balance $V_A = V_B$, the system will remain in a stationary state in which the hole can sit on either redox center with equal probability. Periodically modulating the height of the barrier between the two centers without disturbing the balance cannot change the situation. No energy dissipation, neither a resonance peak in the frequency shift, would be observed in that case. In the experiment, the degree of asymmetry between the two centers differs among different molecules and different probing tips. In general, the spatial location of the resonance has a rainbow-like shape (see Fig. 2g,h) when viewed in the xz plane, i.e. an arc leaning towards one of the redox centers. Sometimes, however, it forms an almost vertical and almost straight line, placed above the molecule roughly in the middle between the two redox units. These instances are presumably those which come the closest to the symmetric arrangement. The vertical line is then typically relatively short compared to the long arcs observed in other cases; the resonance now vanishes above certain height. Here, we may find ourselves close to the situation in which the tip could not change ΔE because of the symmetry between A and B.

We probe the phase shift between the tip oscillations and the charge transfer only indirectly, through the dissipated energy. This signal is unfortunately not sensitive to the direction of the charge transfer.

The theory considers wave-packets, but would a simpler state theory suffice, i.e. just considering if the hole would be either left or right?

Indeed, we have been also analyzing this alternative, which is represented by the all-Marcus model in the manuscript. The all-Marcus model is able to explain the experimental results to some extent, however it completely fails to describe properly the variation of the frequency shift with oscillation amplitude of the probe, as shown on Figs. S13-14 and Fig. 5. We extended the underlying discussion in the manuscript. More detailed discussion of the two model can be found Supplemental Material, section G: *Amplitude dependence of the $\Delta f(z)$ and $E_{diss}(z)$ line shape: experiment and theory.*

In as much tunneling between the two states is included in the theory?

Our electron transfer rate is calculated with second order perturbation theory in the tunnelling element M (this is Marcus theory). Namely, the electron transfer rate k_{ET} between two redox states can be expressed as $k_{ET} \approx |M|^2 e^{-\Delta G/k_B T}$, the electronic coupling $|M|^2$ describes the tunnelling rate between two redox centres. In principle, if the tunnelling element could not be treated perturbatively, more involved theories of electron transfer (including coherent tunnelling) could be applied. We briefly discuss the possibility of models alternative to ours near the end of the section D in the Supplementary text.

The last page contains a lengthy discussion of the model, however, referring to a figure in the SOM. If this discussion is central to the understanding the corresponding figure should appear in the main text.

According to referee's suggestion we moved Fig. S15 into the main text as Figure 5.

Reviewer #3 (Remarks to the Author):

The authors have carried out an extensive study of single electron transfer within a single molecule on an insulating substrate by means of frequency modulated atomic force microscopy under ultra-high vacuum conditions. The molecule aptly chosen consists of two ferrocene redox centers separated by a naphthalene linker. This allows the authors to maintain different oxidation states of Fe in the two centers through selectively extracting 2, 1

or 0 electrons with appropriate gating of the AFM tip. Very interesting results presented in Figure 2 (authors should correct the numbering in the caption of this figure) show a strong signal that appears as a sharp line in both the measured frequency shift and energy dissipated channels in between the two redox centers in the mixed valence state. This signal is missing in the case of the neutral and the fully oxidized (+2h) charge states. The authors attribute the origin of the effect to a continuous switching of the charge between the two redox centers at a certain tip-sample position with respect to the redox centers. Importantly, the effect shows very little dependence on the tip temperature. The rest of the paper is on analysis of these two signals using insights from density functional theory-based calculations and Marcus rate equation.

The observation of single electron transfer is not new, although it is nice to see another example of it. The controlling of the mixed-valent state of the molecule with appropriate gating of the AFM tip is a notable contribution. Perhaps the most important contribution of the work is the temperature independence (almost) of the switching rate which the authors interpret as quantum dissipation. While the authors have carried out detailed analysis of the rate using Marcus theory of classical dissipation and effect of the environment, there is room for more fundamental theoretical work. The authors claim, as a result of extensive analysis, that “the weak temperature dependence of the switching rate is solely due to the non-equilibrium character of the electron transfer”, is reasonable and should stimulate further work using quantum mechanical methods for examination of systems that are far from equilibrium. For example, is this an ultrafast process? Do electron correlations play a role?

We thank the referee for his/her time spent on the evaluation of the manuscript and constructive feedback, which helps us to improve readability of the manuscript.

Authors should proof read the manuscript so careless errors are avoided: labeling of Figure 2, details of Ref. 2, etc.

We fixed the numbering of the labels in the Figure 2. We also changed the arrangement of the panels on Figure 2, see also our response to Reviewer #1. We also fixed the formatting problem with reference 2. Now the whole reference is visible in the manuscript

I recommend publication as the work and its interpretation is novel. It should lead to debate and discussion in the field and trigger more experimental and theoretical work.

REVIEWERS' COMMENTS:

Reviewer #2 (Remarks to the Author):

The revised manuscript has been improved in readability and understanding of the complex content. The authors also replied and answered to all questions raised by the reviewers comprehensively. I agree that the reordering of figures and the movie in the SOM explains nicely the time evolution of the process.

Considering the remark on the isolated nature of this study, I would have hoped for a more detailed reply. I agree that single-molecule studies have much to offer compared to ensemble investigations. The question is, in my view, if the main interest lies on the material/molecule under investigation, where an artificial model system was chosen here, or the method (conductive AFM) which is not entirely new or the interpretation of the observations, which I find the most innovative part of this manuscript. However, in the manuscript the first three references refer to theoretical studies, although one would have expected to learn about intramolecular ET processes in nature. Also, the other references from 4-7 refer mostly to intermolecular charge transfer processes or coupling of molecules to metal electrodes. I would have expected to learn for what particular systems the proposed method or theoretical treatment is valid (time scales of ET processes, spatial extension of the system, etc.). The manuscript is still vague on this which can be also noted from the choice of words ("may") in the conclusions and also rebuttal letter.

This entails also the question on the out-of-equilibrium character of the process. I agree that out-of-equilibrium processes are indeed important but like to stress that this is due to the intrinsic nature of the process whereas here the probe (AFM) is also the cause for the out-of-equilibrium drive of the ET. In this sense, I can think of ET processes that are driven by other external stimuli and then probed and interpreted by AFM measurements. This is different in the presented study.

Having written all that, I can still acknowledge that the presented study and their interpretation is very interesting and deserves publication in Nature Communications. The manuscript presents complex measurement schemes and data and the interpretation is detailed and comprehensive. Therefore, providing some more clue on what are the limitations and where the authors see the impact of their work for others in this field, is very helpful. The current version provides some more information on that but if the authors can find more specific examples, they should include them in the discussion. Readers not entirely familiar with this field will appreciate that very much.

Dear Editor,

We thank all referees for their time spent on the evaluation of our manuscript and for the constructive and valuable feedback. We hope the new version of the manuscript addresses properly the comment of Referee #2.

REVIEWERS' COMMENTS:

Reviewer #2 (Remarks to the Author):

The revised manuscript has been improved in readability and understanding of the complex content. The authors also replied and answered to all questions raised by the reviewers comprehensively. I agree that the reordering of figures and the movie in the SOM explains nicely the time evolution of the process.

Considering the remark on the isolated nature of this study, I would have hoped for a more detailed reply. I agree that single-molecule studies have much to offer compared to ensemble investigations. The question is, in my view, if the main interest lies on the material/molecule under investigation, where an artificial model system was chosen here, or the method (conductive AFM) which is not entirely new or the interpretation of the observations, which I find the most innovative part of this manuscript. However, in the manuscript the first three references refer to theoretical studies, although one would have expected to learn about intramolecular ET processes in nature. Also, the other references from 4-7 refer mostly to intermolecular charge transfer processes or coupling of molecules to metal electrodes. I would have expected to learn for what particular systems the proposed method or theoretical treatment is valid (time scales of ET processes, spatial extension of the system, etc.). The manuscript is still vague on this which can be also noted from the choice of words ("may") in the conclusions and also rebuttal letter.

This entails also the question on the out-of-equilibrium character of the process. I agree that out-of-equilibrium processes are indeed important but like to stress that this is due to the intrinsic nature of the process whereas here the probe (AFM) is also the cause for the out-of-equilibrium drive of the ET. In this sense, I can think of ET processes that are driven by other external stimuli and then probed and interpreted by AFM measurements. This is different in the presented study.

Having written all that, I can still acknowledge that the presented study and their interpretation is very interesting and deserves publication in Nature Communications. The manuscript presents complex measurement schemes and data and the interpretation is detailed and comprehensive. Therefore, providing some more clue on what are the limitations and where the authors see the impact of their work for others in this field, is very helpful. The current version provides some more information on that but if the authors can find more specific examples, they should include them in the discussion. Readers not entirely familiar with this field will appreciate that very much.

We appreciate the referee critical assessment of the limitations of our method and its implications to general studies of the electron transfer processes in diverse systems. We try to address these comments in the new version of the manuscript, extending substantially Introduction part and also modifying the Discussion part.

On behalf of all authors with kind regards,

Ass. Prof. Pavel Jelínek
Institute of Physics of the AS CR
Cukrovarnická 10
Prague 6
CZ-162 00
Czech Republic
Email: jelinekp@fzu.cz
Tel.: +420 220 318 430
Fax: +420 233 343 184
[www: nanosurf.fzu.cz](http://www.nanosurf.fzu.cz)